



**Combined diurnal variations of discharge and hydrochemistry of the Isunnguata Sermia outlet of the**
**Greenland Ice Sheet give insight on subglacial conditions**
Joseph Graly, Joel Harrington, Neil Humphrey
University of Wyoming
**Abstract**
In order to examine daily cycles in meltwater routing and storage in the Isunnguata Sermia
outlet of the Greenland Ice Sheet, variation in outlet stream discharge and in major element
hydrochemistry were assessed over a six day period in July, 2013. Discharge was assessed from hourly
photography of the outlet from multiple vantages, including where mid-stream naled ice provided a
natural gauge. pH, electrical conductivity, suspended sediment, and alkalinity were measured in samples
of stream water collected every three hours. Element and ion concentrations were subsequently
measured in a laboratory setting.
Photography and stream observations reveal that although river width and stage have only
slight diurnal variation, there are large changes in discharge shown in the portion of the width
characterized by standing waves and fast flow. Width of this active channel approximately doubles over
a diurnal cycle. Together with changes in flow over the naled, these features allow an observationally
based relative record of stream discharge in this unconstrained alluvial setting. Peaks in discharge were
offset by 3-7 hours from peak melt of the interior ice surface.
Concentration of dissolved solutes follows a sinusoidal diurnal cycle, except for large and
variable increases in dissolved solutes during the stream's waning flow. Diurnal changes in solute
concentration average 31% of the base value. Diurnal solute concentration minima and maxima lag peak
and minimum stream discharge by 3-6 hours.
This phase shift between discharge and solute concentration suggests that during high flow,
water is either encountering more rock material or is stored in longer contact with rock material. We



suggest that expansion of a distributed subglacial hydrologic network into seldom accessed regions
during high flow could account for these phenomena, and for a spike of partial silicate reaction products
during waning flow, which itself suggests a pressure threshold-triggered release of stored water.

**1. Introduction**

Dissolved load in glacial outlet streams has long been employed as a metric for assessing water-

rock interactions occurring beneath glaciers and ice sheets. Glacierized basins have comparable
dissolved loads to non-glacial rivers, but are enriched in mobile cations and depleted in Si (Anderson et
al., 1997). Chemistry of glacial water suggests that observed solute concentrations are reached due to
presence of reactive accessory minerals and fresh mineral surfaces in glacial sediments (Drever and
Hurcomb, 1986). Dissolved load is therefore linked to physical erosion in subglacial environments
(Anderson, 2005). Dissolved load is also indicative of the degree to which atmospheric gases have been
sequestered by chemical processes in the subglacial environment (Hodson et al., 2000).

Diurnal variation of solute concentration is a potential indicator of meltwater routing and

storage. Solute concentration is controlled by total water-rock contact during water residence time in
the subglacial environment and by reactivity of minerals contacted by the water. In particular, two end
member cases are expected: if dilution produces an inverse relationship between discharge and solute
concentration, minimal changes in water-rock interaction over time are suggested, whereas if increased
discharge is coupled to increased solute concentration, diurnal changes in processes of water-rock
interaction or storage are suggested.

Studies of alpine glaciers have typically found solute concentration and discharge to vary

inversely, with rising discharges corresponding to falling concentrations of dissolved solutes (Collins,
1995; Hindshaw et al., 2011; Tranter et al., 1993). Ions produced by saturation limited reactions, such as
calcite dissolution, can show increased load with discharge, but typically with diminished concentration



per water volume (Mitchell and Brown, 2007). Elements that are limited by factors such as the rate of
sorption/desorption will have constant flux levels and will only be diluted by increased water flow
(Mitchell and Brown, 2007). Resultantly, correlations between discharge and dissolved load are typically
weak (Collins and MacDonald, 2004). These dilution relationships have been attributed to the
dominance of conduit flow in alpine environments. In cases where subglacial water is confined to fixed
conduits, increased water flow will expand the size of these conduits and the speed of through-flow but
will have a minimal impact on the area of water-bed contact (Nye, 1976; Röthlisberger, 1972).

Studies of larger glacial systems suggest more complex water-rock interactions. In the outlet of a

large glacierized basin in SE Alaska, increases in dissolved load lag spikes in discharge by several days
(Anderson et al., 2003). Anderson and others attribute this to storage of water in a distributed system
only released during the waning stages of flow. In distributed or linked-cavity flow, increased discharge
allows flowing water to spread out across the glacier's bed and thereby increase the area of water-bed
contact (Humphrey, 1987; Kamb, 1987). Time series data from the Watson River, near Kangerlussuaq,
West Greenland, show out of phase variation in discharge and solute concentration, with maximum
daily solute concentrations occurring as discharge is rising and minima occurring as discharge is falling
(Yde et al., 2014). However, on the scale of the melt season as a whole, Yde and others find a strong
inverse correlation between discharge and solute concentration, which they attribute to conduits
carrying a substantially higher portion of the meltwater flow than the distributed subglacial system. Lags
between minimum discharge and peak solute concentrations are also observed at Tsanfleuron Glacier,
Swiss Alps (Zeng et al., 2012).

Seven measurements of dissolved solute chemistry taken from samples collected over the

course of 2 days in 2011 at the terminus of a major west Greenland outlet glacier potentially show a
direct relationship between solute concentration and discharge (Graly et al., 2014; Landowski, 2012). In
this limited time series, solute concentration appears to peak at midafternoon, while discharge is high,



and be minimal in the early morning hours, with total variation of <20%. To further investigate whether
a direct relationship between discharge and solute concentration exists in the Isunnguata Sermia outlet
of the western Greenland Ice Sheet, we returned to the same site for a six day period of the summer of
2013, collecting 8 samples per day for chemical analysis.

**2. Field site**

Water samples were collected from the terminus of Isunnguata Sermia, a major land-

terminating outlet of the western Greenland Ice Sheet (Figure 1). The outlet glacier Isunnguata Sermia
occupies a deeply cut glacial valley, with surrounding hilltops >400 m above sea level. Deep, glacially-
carved trenches continue under the ice sheet for more than 20 km into interior, with ice depths of
greater than 1,000 m (Jezek et al., 2013). The Isunnugata Sermia outlet has a catchment that
encompasses >2,400 km$^2$ of the ablation zone, making it one of the largest regional subglacial drainage
basins (Palmer et al., 2011). Regional geology consists of primarily of Paleoproterozoic gneisses and
granitoids (van Gool et al., 2002). The Isunnguata Sermia outlet is located 25 km from the town of
Kangerlussuaq. It is the next outlet to the north of the Russell Glacier and feeds a separate glacial river
system from the Watson River.

Water emerges from a single location on the south side of Isunnguata Sermia's terminus front

~30m above sea-level (Figure 1) and traverses a broad, >100 km long sandur to the fjord. Discharge of
pressurized subglacial water creates a large upwelling capable of expelling water multiple meters into
the air and, although no fully quantitative measurement could be made, peak discharge was estimated
to exceed 500 m$^3$s$^{-1}$. Ice-cored moraines and frozen outwash shape the course of outlet waters. On the
sandur, frozen outwash channels the water into a single thread, although the large sediment load
creates rapidly changing channel and bed geometry. Near the terminus, the frozen outwash of the
sandur is elevated ~2-4 meters above the discharging stream. The main stream is also fed by minor ice



surface melt streams. A small stream that runs along the south lateral margin of the glacier joins the
main terminal outlet stream just below the primary upwelling site.
This work was performed in the context of several other studies of the Isunnguata Sermia
terminus. Hot water boreholes along a transect from the outlet to 40 km in the interior have provided
data regarding water pressure (Meierbachtol et al., 2013), ice temperature (Harrington et al., 2015),
subglacial water chemistry (Graly et al., 2014), and mass balance between subglacial sediment and rock
(Graly et al., 2016). More limited datasets from Isunnguata Sermia's terminal and lateral outlets were
reported for the 2010, 2011, and 2012 seasons (Graly et al., 2014). The work reported here is based on
samples and data collected over a 6 day period from July 16th to July 21st, 2013.

**3. Methods**
**3.1 Water Sampling**
Water sampling began at 8:00 hours local time on July 16th, 2013 and continued in 3 hour
increments through 20:00 hours on July 21st, 2013. Samples were collected by lowering a liter bottle
attached to a pole into discharging waters within 400 m of the subglacial upwelling. During July 16th and
July 17th, samples were collected from the south bank of the outlet stream from the banks at the
beginning of the outwash plain (Figure 1). During July 17th, the main course of the river shifted so that
location had diminished flow and an emerging bank channeled waters from the lateral side stream to
the location. Commencing at 14:00 hours on July 18th, sampling was relocated above the lateral side
stream on the banks of the terminal moraine (Figure 1). The sampling location was not subsequently
changed. Excepting periods where the emerging bank channeled lateral stream water to the first
location, both locations sampled water from the main subglacial outlet and should produce comparable
results.



Upon collection, 125 ml of water sample were pumped through 0.1 μm nylon filters, with
filtered water and filter papers saved for laboratory analyses. A colorimetric alkalinity test, a
conductivity measurement, and a pH measurement were performed on the remaining unfiltered
sample. Alkalinity tests were performed with a Hach Model AL-AP alkalinity test kit. Results of field
alkalinity tests were only accurate to 25 μM. Alkalinity was therefore also calculated by charge balance
from the other measured ions. pH measurements and conductivity measurements were performed with
Beckman-Coulter Φ460 multi-parameter meter. pH was measured using a low ionic strength probe.
Subsequent water analyses were performed in the University of Wyoming Aqueous
Geochemistry Lab. Concentrations of Si, Ca, Mg, Na, and K were measured on a Perkin Elmer Elan 6000
inductively coupled plasma quadrupole mass spectrometer (ICP-MS). Concentrations of $SO_4^{2-}$, $Cl^-$, $NO_3^-$,
and $Fl^-$ were measured on a Dionex ICS 500 ion chromatograph. The filter papers were dried and
weighed to assess suspended load.

**3.2 Discharge**
Discharge measurements of the outlet were difficult. There is no exposed bedrock near the
stream to act either as an elevation reference or to stabilize the river bed. Obtaining accurate cross
profiles of the stream was prohibitively dangerous, with high flows, collapsing banks, and a considerable
flux of mobile ice blocks. Attempts to install a stage pole were frustrated by considerable stage variation
over time associated with cutting and filling of the river bed and banks. Once the stream opens out from
its restriction by remnant glacier ice near the upwelling, stage is poorly correlated with discharge. The
stream instead scours sediment during the rising limb and deposits it on the trailing limb of the daily
hydrograph. It was decided to assess only relative discharge. This was aided by repeat photography from
two fixed locations.



Hourly stream photography began at 10:00 hours on July 18[th] and continuing through 20:00
hours on July 21[st]. From one vantage point, the central upwelling of subglacial water was photographed
from the south, as it poured out from around the moraine. This vantage captured a ~1 meter high, mid-
channel naled formed from freezing of outlet waters during winter months. The naled was variably
covered or exposed as discharge varied and acted as a stream gauge in this respect. This portion of the
stream is restricted by frozen sediment and stream height is controlled by discharge.
A second vantage, from a rise above the south bank, captured a ~200m long stretch of the
outlet stream. In this portion of the stream, increased discharge caused scour and expansion of the
stream's active channel. Photography allowed assessment of relative active channel width. Large waves
and faster velocities are confined to this active channel, allowing fairly unambiguous, though qualitative,
determination of which portions of an overhead photograph comprise the active channel. The distance
between the upstream end of a persistent, mid-stream point bar and a distinct feature on the south
shore was measured on each photograph (Figure 1). The length of the portion of this transect
characterized by large waves and flow features was also measured, allowing for the calculation of the
percentage of the stream width contained by the active channel. The second vantage also allowed
assessment of flow state and Froude number from the presence of features such as standing waves.
During the first two days of the sampling period, stream surface velocity was measured by
repeated timing the motion of floating ice and other stream surface features down a 100 m section of
the stream. Measurements were taken during morning, afternoon, and evening stages to assess
variation in velocity associated with high and low flow.

**3.3 Interior Surface Melt**
In order to compare variation in terminus discharge to melt in the surface interior, we are also
including discharge measurements from an interior ice sheet surface stream. The stream was gauged



during the summer of 2012, so data are not directly comparable to the measurements collected in 2013.
However inasmuch as interior melt is primarily controlled by insolation, the stream's variation likely
represents a typical pattern for the timing and scaling of diurnal summer surface melt fluctuations.

The surface stream was located at 67.2˚N and 49.8˚E, ~25 km from the terminal outlet. Stream

height was gauged with a calibrated pole drilled into ice. Surface velocity was measured by timing
floating ice along a course of known distance. Cross-sectional area was directly measured in the region
where the gauge was emplaced and calibrated to gauge height. Transect slope was measured by pole
and automatic level. Six measurements of surface velocity used to calculate an average Manning
coefficient from the measured slope and hydraulic radius of the stream. Discharge was then calculated
from change in gage height. Stage height was measured every half hour or hour for a period from 11:30
6/18/12 to 20:00 6/21/12. During June 18[th], 19[th], and 20[th], sunny weather predominated; June 21[st] had
rainy, cooler weather.

**4. Results**
**4.1 Discharge**

Over the four days during which repeat photographic observations were made, photographs of

the naled show consistent minima at 8:00 hours, with the naled is mostly exposed, and a small volume
of water overtopping a portion of the ice body (Figure 2). During the first two days of observation, the
naled was completely covered by flowing water from 19:00 to 0:00 hours. On the third day it was
covered from 16:00 hours, and remained covered for the remainder of the study period.

Maximum discharge is harder to determine from observations of the naled alone. Once the

naled is completely covered in water, visual interpretation of maximum flow is ambiguous. Some
discrimination can be made based on height of the covered naled feature compared to the surrounding
waves and angle at which the water pours over the naled (greater flows overtop the naled at a lower



angle). From these features, maximum stream flow appears to occur at 21:00 hours on July 18[th] and
19[th], and 20:00 hours on July 20[th].
Standing waves are observed at all flows (Figure 2), although substantial differences in wave and
surface morphology were noted during waxing and waning phases, with rougher water in waning flow
and smoother water in waxing flow. The roughness change may represent a change in the sediment load
of the river between the erosive waxing stage and the depositional waning stage. The persistence of
standing waves implies near critical flow conditions, or a Froude number approximately 1, for the entire
study period. Measurements of stream velocity showed surface speeds of 2.86 ± 0.12 m/s ($2\sigma$, n=6).
Variation in velocity between morning and evening stages was within measurement error. Based on
calculations from a Froude number of 1, stream depths of 0.5 - 0.9 m are suggested; these depth
estimates were supported by observing ice blocks rolling or bouncing down the flow. The lack of
relationship between stage and discharge and velocity has been noted before in sediment laden glacial
rivers (Humphrey and Raymond, 1994).
Since neither stream velocity nor depth change with discharge, variations in discharge are
accommodated by changes in the width of the active channel. Wide areas of shallow slow water
remained present during low flows and the total surface area of the stream remained approximately
constant. Pole probing of these shallow areas suggests 10 to 20 cm depths. Because the active channel
has an order of magnitude greater discharge per transect meter than the marginal areas and changes in
active channel water velocity were not observed, cross-sectional area of the active channel is the
primary control on discharge.
Assessment of the active channel width shows substantial differences between morning hours
(~5:00-10:00), where 20-30% of the stream is comprised of active channel characteristics, and late
afternoon / evening hours (~18:00-0:00) where >40% of the stream is comprised of active channel





characteristics. These observations are generally consistent with assessments of the height of water
pouring over the naled (Figure 3).

**4.2 Interior Surface Stream**

The calculated Manning coefficient for the interior stream was 0.0117 ± 0.0018 (2σ). Discharge

measured in the interior ice surface stream varied by as much as an order of magnitude during the
course of diurnal cycles, with low values as small as 0.3 $m^3s^{-1}$ and high values greater than 3.5 $m^3s^{-1}$
(Figure 3). Minimum stage heights consistently occurred around 4:00 hours. Maximum stage heights
consistently occurred at 14:00 or 15:00 hours. These data contrast with our observations of water
pouring over the naled. The naled minimum occurred approximately 4 hours later than minimum of
surface melt. The naled maximum occurred approximately 6 hours later than maximum surface melt.
This delay is representative of the integration of the travel time delays from the entire glacier
catchment.

**4.3 Water Analyses**

Sampled waters are generally chemically dilute, with 292±50 micromoles per liter dissolved

solutes (Table 1). Ca is the dominant cation, followed by Na, K, and Mg (Figure 4). Mg abundances are an
order of magnitude lower than the other major cations. Dissolved Si occurs at comparable abundance to
Na. Standard deviations of the mass spectrometer measurements were <1% of the measured value.
Bicarbonate (measured as alkalinity) is the dominant anion. $SO_4^{2-}$ and $Cl^-$ are detected in all samples, but
occur at an order of magnitude lower concentration. Trace amounts of $NO_3^-$ and $Fl^-$ were detected in
some samples, at values an order of magnitude below $SO_4^{2-}$ and $Cl^-$ concentrations (Figure 4). On
average, field alkalinity measurements exceed the alkalinity estimates from charge balance by 25 ± 14
μM (2σ). Some over-measurement in the field titration is expected, as the value is recorded at the level



where the color tracer disappears (and therefore is a maximum compared to previous drop). Charge
imbalance may also result from absorption of $H^+$ particles to suspended sediment in unfiltered water.
Field electrical conductivity measurements show similar results to the laboratory analyses ($p<0.0001$)
(Figure 4). Suspended sediment concentration does not show a consistent correlation or anti-correlation
with dissolved load (Figure 4).

Relative abundances of cation species is comparable to measurements taken at the Isunnguata

Sermia terminus in the summer of 2011 (Graly et al., 2014). The $SO_4^{2-}$/alkalinity ratios are diminished
compared to those measured in 2011, but are comparable to those found in samples collected in 2010
and 2012. The concentrations of suspended sediment are similar to those observed at nearby Leverett
Glacier during the summer of 2010 (Cowton et al., 2012).

When normalized to average concentration, magnitude and timing of cation and silica

concentration variation is highly consistent between species over time (Figure 5). Covariation of all
cation and Si species is statistically significant with $p<0.05$. Covariations of K-Mg, K-Si, and Na-Si have p-
values ranging from 0.01 to 0.05; all others are <0.0001. All cations and silica concentrations follow a
diurnal pattern with higher concentrations during morning and early afternoon hours and significantly
lower concentrations during later afternoon and evening hours. In several of the studied cycles, large
changes in total concentration are limited to the 20:00 and 23:00 hours samples, which are substantially
lower than the other samples collected throughout the day.

There are two major deviations from the diurnal pattern. The 11:00, 14:00, 17:00, and 20:00

samples from July 17[th] have substantially lower concentrations than would be otherwise suggested by
diurnal fluctuations observed elsewhere in the record. This corresponds with the period during which an
emerging bank partially separated site 1 from the main channel allowing a surface-fed side stream to
substantially dilute the water.





At 2:00 on July 20$^{th}$, there is a >60% spike in total concentration of all cations. Similar, but
smaller magnitude spikes are also present in the 2:00 samples of July 19$^{th}$ and 21$^{st}$ and the July 16$^{th}$
23:00 sample. The most clearly expressed of these spikes (July 20$^{th}$ and 21$^{st}$) are substantially more
expressed in Na and K concentrations than in Ca, Mg, or Si. During the July 21$^{st}$ spike, the spike in Mg
and Si appears to proceed the spike in Ca, Na and K in that it appears during the previous sampling
period. Large variability in the magnitude of these spikes suggests that the 3-hour sampling schedule
was insufficiently frequent to characterize them entirely.
Anions generally follow similar patterns, but with greater variability (Figure 4). In particular, Cl
does not co-vary with other ions toward the end of the record.  The spike on July 18$^{th}$ coincides with a
drop in SO$_4$ concentration; the spike on the 20$^{th}$ coincides with a drop in Cl concentration. SO$_4$
concentrations generally only minimally increase during the spikes.
Excluding these anomalies, the highest concentration of dissolved solids occurs at 11:00 on July
16$^{th}$, 19$^{th}$, 20$^{th}$, and 21$^{st}$ (Figure 5). On July 17$^{th}$ and 18$^{th}$, the 11:00 sample was likely diluted by the side
stream (which was a significant component of flow to site 1 during that period). Concentration minima
are reached at 23:00 hours on July 17$^{th}$ through 20$^{th}$. On July 16$^{th}$, the minima occurs in the 20:00
sample. The size of the diurnal variation varies from 22% to 49% of the lowest value, with an average
daily range of 31 ± 9%.

**5. Discussion**
*5.1 Discharge and Outlet Stream Observations*
Observations from oblique photography suggest large diurnal changes in discharge. Width of the
active channel, with deeper faster water, approximately doubles in the course of the day (Figure 3b). An
approximate doubling of discharge is also suggested by observations of the midstream naled. The naled
is of comparable scale to the depth of the stream (both order of 1 meter). Its exposure during low flow



and burial during high flow suggests a change in stage comparable to its height. At the naled site,
increased width of active channel flow is restricted by ice. Increases in flow height at the naled location
are therefore approximately equivalent to increases in active channel width downstream.

During high flows, diurnal increases in discharge of up to 50% of base value are observed in the

Watson River at Kangerlussuaq, where a bridge over a narrow gorge has allowed for the construction of
a reliable gauge (Hasholt et al., 2013). As the Watson River is 20-30 km from its glacial outlet sources
and integrates several independent glacial outlet streams, these diurnal cycles are likely muted
compared to their expression at the ice margin. Larger diurnal changes are therefore expected directly
at glacier outlet termini. Contrastingly, Smith and others (2015) found minimal diurnal variation in
discharge at Isunnguata Sermia terminus. However, as Smith and others estimated discharge based
solely on the surface area of the outlet stream water, their analysis missed the variation in the width of
the active channel and height of its flow over static features that we present.

*5.2 Diurnal Changes in Solute Flux*

The critical observation is that variability in the dissolved solute concentrations cannot be

explained by dilution alone. First, the scale of discharge variation is substantially larger than variation in
concentration of dissolved solutes. Approximate width of the main channel doubles during diurnal
cycles, while concentrations of dissolved solutes only change by an average of 21-40% (Figure 5).
Secondly, maximum and minimum solute concentrations are offset from minimum and maximum
discharge. Such lags imply periods where solute concentration is increasing even as discharge rises and
periods where solute concentration is falling even as discharge falls.

Periods of in-phase changes between discharge and solute concentration suggest that increased

water flow is either stimulating increased water-rock interaction or allowing for release of stored water
(that has developed higher solute concentrations over longer residence times). While the single



upwelling structure of the terminus of Isunnguata Sermia implies local channelized flow, observations of
water pressures at interior sites (Meierbachtol et al., 2013) and hydrologic theory for low ice surface
slopes (Werder et al., 2013) both suggest that much of the catchment interior has a linked-cavity flow
system. Linked cavity systems would allow for expansion of the basal hydrological system and flushing of
long water residence time regions under high flow conditions.

Sudden increases in solute concentration during waning flow suggest that discharge from

subglacial regions with a high concentration of dissolved solutes is triggered when a threshold is
reached. Multiple triggering mechanisms are plausible. Modeling of subglacial water pressures suggests
that near the ice sheet margin, water flows from conduits to the distributed cavity system at high
conduit water pressures and back to conduits at low pressures (Meierbachtol et al., 2013). Solute
concentration spikes result from the crossing of a pressure threshold allowing water stored during high
flow to suddenly enter the glacial outlet system.

Solute concentration spikes might also be explained by creep closure of linked cavities that

opened during high flow and expulsion of remaining solute-concentrated water. Anderson and others
(2003) proposed a similar creep closure mechanism to explain increases in solute concentration during
waning flow that occurred on a multiday scale in a mountain glacier setting. Following the Glenn-Nye
relation, the rate of creep closure of ice scales to approximately the third power of effective pressure
(Cuffey and Paterson, 2010). Differences in timing of these effects between ice sheets and mountain
glaciers can therefore be explained by differences in ice thickness.

Relative dominance of Na and K in these spikes is consistent with water-rock interactions

occurring only over a limited time, such that cation exchange occurred on fresh feldspar and mica
surfaces but complete silicate dissolution and clay precipitation did not (Blum and Stillings, 1995; Graly
et al., 2014). Contrastingly, constituents associated with weathering of reactive accessory minerals such
as pyrite and calcite (especially $SO_4$) are minimally expressed. This implies that the spikes' composition

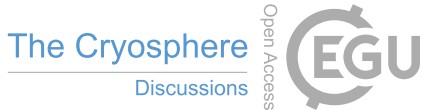

reflects waters that have rapidly passed through reactive sediment that is depleted of accessory
minerals. Such accessory mineral depletion can occur if sediment residence time in the subglacial
system is sufficiently long (Graly et al., 2014). Sampling of sediment beneath ice boreholes has shown
the greatest chemical depletion in portions of the ice sheet most likely to be influenced by distributed
flow (Graly et al., 2016). This suggests that the spike of chemical solutes comes from water that has
temporarily entered regions of distributed flow as a part of a diurnal cycle.

**6. Conclusions**

A semi-quantitative discharge record can be constructed through hourly photographic
monitoring of the static and dynamic features of a large, sediment laden glacial outlet stream. These
assessments show large diurnal changes in discharge over the six day study period at the Isunnguata
Sermia outlet of the Greenland Ice Sheet (c.f. Smith et al., 2015). Simultaneously collected chemical
measurements show substantially smaller fluctuation in dissolved load; thus this Greenland outlet
glacier does not show the discharge-driven dilution of solute concentration that is common in smaller
ice masses. Periods where dissolved solute concentration increase and decrease along with discharge,
and abrupt and variable increases in solute concentration during waning flow imply that significant
contributions to the solute load is made by changes to the routing and storage of meltwater in the
subglacial system over the course of the day. In particular, these results indicate considerable exchange
of water diurnally between the conduit and linked cavity drainage systems, as well as implying threshold
pressure conditions for these exchanges.

*Acknowledgements.*This work would not have been possible without funding from the Greenland
Analogue Project (SKB, Posiva, NWMO) and NSF grant ARC-0909122. Janet Dewey assisted with
laboratory analyses.




**Cited References**
Anderson, S.P., 2005. Glaciers show direct linkage between erosion rate and chemical weathering fluxes.

Geomorphology 67, 147-157.

Anderson, S.P., Drever, J.I., Humphrey, N.F., 1997. Chemical weathering in glacial environments. Geology

25, 399-402.

Anderson, S.P., Longacre, S.A., Kraal, E.R., 2003. Patterns of water chemistry and discharge in the

glacier-fed Kennicott River, Alaska: Evidence of subglacial water storage cycles. Chemical Geology

202, 297-312.

Blum, A.E., Stillings, L.L., 1995. Feldspar dissolution kinetics, in: White, A.F., Brantley, S.L. (Eds.),

Chemical Weathering Rates of Silicate Minerals. Mineralogical Soc Amer, Chantilly, pp. 291-351.

Collins, D., 1995. Dissolution kinetics, transit times through subglacial hydrological pathways and diurnal

vatiations of solute content of meltwaters draining from an alpine glacier. Hydrological Processes 9,

897-910.

Collins, D.N., MacDonald, O.G., 2004. Year-to-year variability of solute flux in meltwaters draining from a

highly-glacierised basin. Nordic Hydrology 35, 359-367.

Cowton, T., Nienow, P., Bartholomew, I., Sole, A., Mair, D., 2012. Rapid erosion beneath the Greenland

ice sheet. Geology 40, 343-346.

Cuffey, K.M., Paterson, W.S.B., 2010. The Physics of Glaciers, 4th ed.
Drever, J.I., Hurcomb, D.R., 1986. Neutralization of atmospheric acidity by chemical weathering in an

alpine drainage basin in the North Cascade Mountains. Geology 14, 221-224.

Graly, J.A., Humphrey, N.F., Harper, J.T., 2016. Chemical depletion of sediment under the Greenland Ice

Sheet. Earth Surface Processes and Landforms In Press.



Graly, J.A., Humphrey, N.F., Landowski, C.M., Harper, J.T., 2014. Chemical weathering under the

Greenland Ice Sheet. Geology 42, 551-554.

Harrington, J., Humphrey, N.F., Harper, J.T., 2015. Temperature distribution and thermal anomalies

along a flowline of the Greenland Ice Sheet. Annals of Glaciology 56(70), 98-104.

Hasholt, B., Mikkelsen, A.B., Nielsen, M.H., Larsen, M.A.D., 2013. Observations of runoff and sediment

and dissolved loads from the Greenland Ice Sheet at Kangerlussuaq, West Greenland, 2007 to 2010.

Zietschrift fur Geomorphologie 57, sup. 2, 3-27.

Hindshaw, R.S., Tipper, E.T., Reynolds, B.C., Lemarchand, E., Wiederhold, J.G., Magnusson, J.,

Bernasconi, S.M., Kretzschmar, R., Bourdon, B., 2011. Hydrological control of stream water

chemistry in a glacial catchment (Damma Glacier, Switzerland). Chemical Geology 285, 215-230.

Hodson, A., Tranter, M., Vatne, G., 2000. Contemporary rate of chemical denudation and atmospheric

$CO_2$ sequestration in glacier basins: An arctic perspective. Earth Surface Processes and Landforms

25, 1447-1471.

Humphrey, N.F., 1987. Coupling between water pressure and basal sliding in a linked-cavity hydraulic

system, The Physical Basis of Ice Sheet Modelling. IAHS Publ. No. 170, pp. 105-118.

Humphrey, N.F., Raymond, C.F., 1994. Hydrology, erosion and sediment production in a surging glacier:

Variegated Glacier, Alaska, 1982-83. Journal of Glaciology 40, 539-552.

Jezek, K., Wu, X., Paden, J., Leuschen, C., 2013. Radar mapping of Isunnguata Sermia, Greenland. Journal

of Glaciology 59, 1135-1147.

Kamb, B., 1987. Glacier surge mechansim based on linked cavity configuration of the basal water conduit

system. Journal of Geophysical Research 92, 9083-9100.

Landowski, C., 2012. Geochemistry and subglacial hydrology of the West Greenland Ice Sheet, MS

Thesis, Geology and Geophysics. University of Wyoming.



Meierbachtol, T., Harper, J., Humphrey, N., 2013. Basal drainage system response to increasing surface
melt on the Greenland Ice Sheet. Science 341, 777-779.
Mitchell, A.C., Brown, G.H., 2007. Diurnal hydrological - physicochemical controls and sampling methods
for minor and trace elements in an Alpine glacial hydrological system. Journal of Hydrology 332,

123-143.

Nye, J.F., 1976. Water flow in glaciers: Jokulhlaups, tunnels, and veins. Journal of Glaciology 17, 181-207.
Palmer, S., Shepherd, A., Nienow, P., Joughin, I., 2011. Seasonal speedup of the Greenland Ice Sheet
linked to routing of surface water. Earth and Planetary Science Letters 302, 423-428.
Röthlisberger, H., 1972. Water Pressure in Intra- and Subglacial Channels. Journal of Glaciology 11, 177-

203.

Smith, L.C., Chu, V.W., Yang, K., Gleason, C.J., Pitcher, L.H., Rennermalm, A.K., Legleiter, C.J., Behar, A.E.,
Overstreet, B.T., Moustafa, S.E., Tedesco, M., Forster, R.R., LeWinter, A.L., Finnegan, D.C., Sheng, Y.,
Balog, J., 2015. Efficient meltwater drainage through supraglacial streams and rivers on the
southwest Greenland ice sheet. Proceedings of the National Academy of Sciences of the United
States of America 112, 1001-1006.
Tranter, M., Brown, G., Raiswell, R., Sharp, M., Gurnell, A., 1993. A conceptual model of solute aquisition
by Alpine glacial meltwaters. Journal of Glaciology 39, 573-581.
van Gool, J.A.M., Connelly, J.N., Marker, M., Mengel, F.C., 2002. The Nagssugtoqidian Orogen of West
Greenland: Tectonic evolution and regional correlations from a West Greenland perspective.
Canadian Journal of Earth Science 39, 665-686.
Werder, M.A., Hewit, I.J., Schoof, C.G., Flowers, G.E., 2013. Modeling channelized and distributed
subglacial drainage in two dimensions. Journal of Geophysical Research: Earth Surface 118, 1-19.



Yde, J.C., Knudsen, N.T., Hasholt, B., Mikkelsen, A.B., 2014. Meltwater chemistry and solute export from

a Greenland Ice Sheet catchment, Watson River, West Greenland. Journal of Hydrology 519, 2165-

2179.

Zeng, C., Gremaud, V., Zeng, H., Liu, Z., Goldscheider, N., 2012. Temperature-driven meltwater

production and hydrochemical variations at a glaciated alpine karst aquifer: implication for the

atmospheric $CO_2$ sink under global warming. Evironmental Earth Science 65, 2285-2297.




**Figures:**

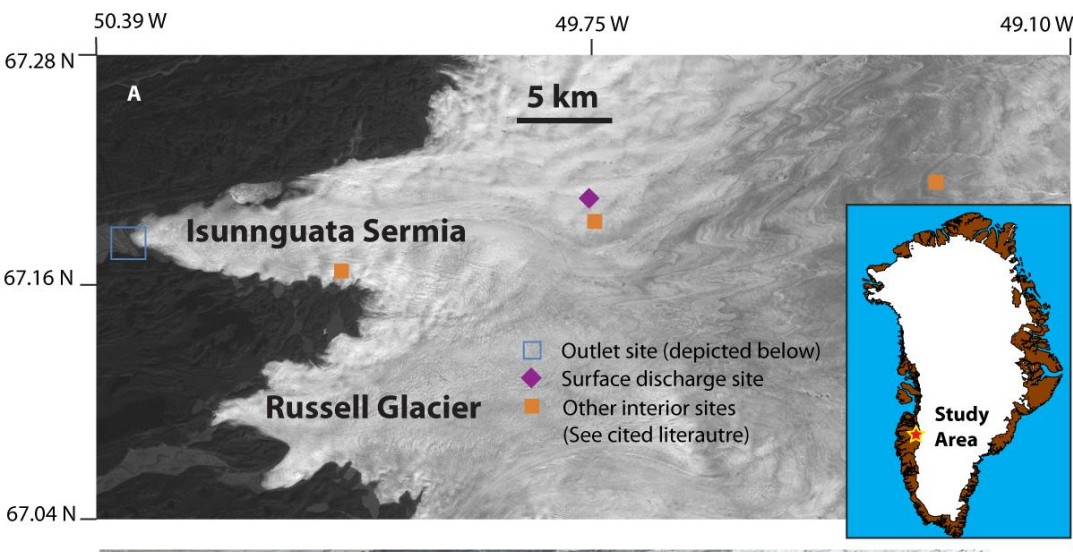

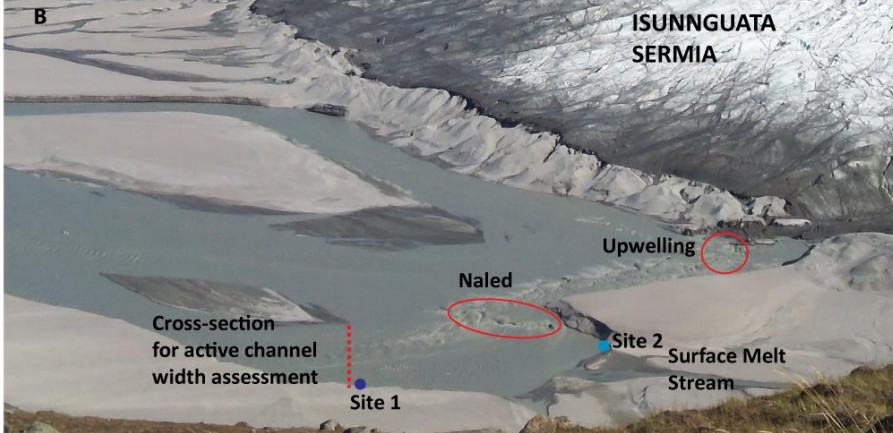


**Figure 1.** A) Location of the study area. B) Overhead photograph of the study area (taken 7/22/13) with
important sampling and observational features.  Samples were collected at site 1 from 10:00 hrs on
7/16/13 through 11:00 hours on 7/18/13. Samples were collected at site 2 from 14:00 hrs on 7/18/13
through 20:00 hrs on 7/22/13.  Beginning at 10:00 hrs on 7/18/13, hourly photographs on the labeled
naled and ~100 m cross-section were taken.



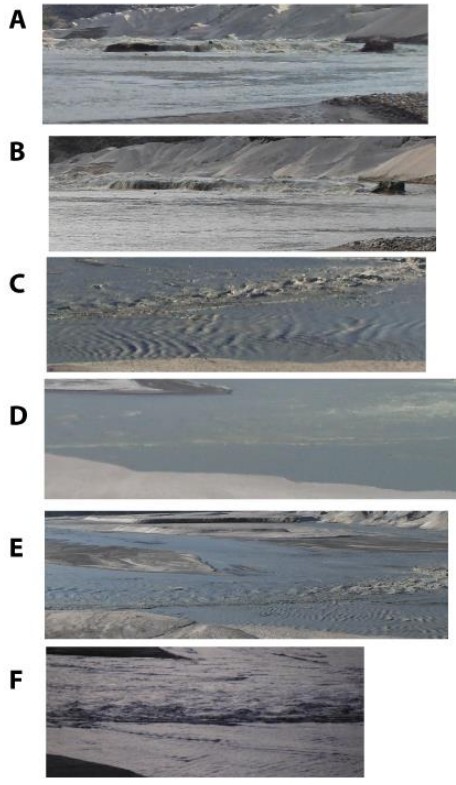


**Figure 2.** Photographs of typical flow patterns in the Isunnguata Sermia outlet.  A) Midstream naled

exposed at during low flow (8:00).  B) Midstream naled covered during high flow (21:00). C-F show flow

during low (8:00), waxing (14:00), high (21:00) and waning (0:00) stages. Waxing and waning stages

show different wave morphology but maintain standing wave features.


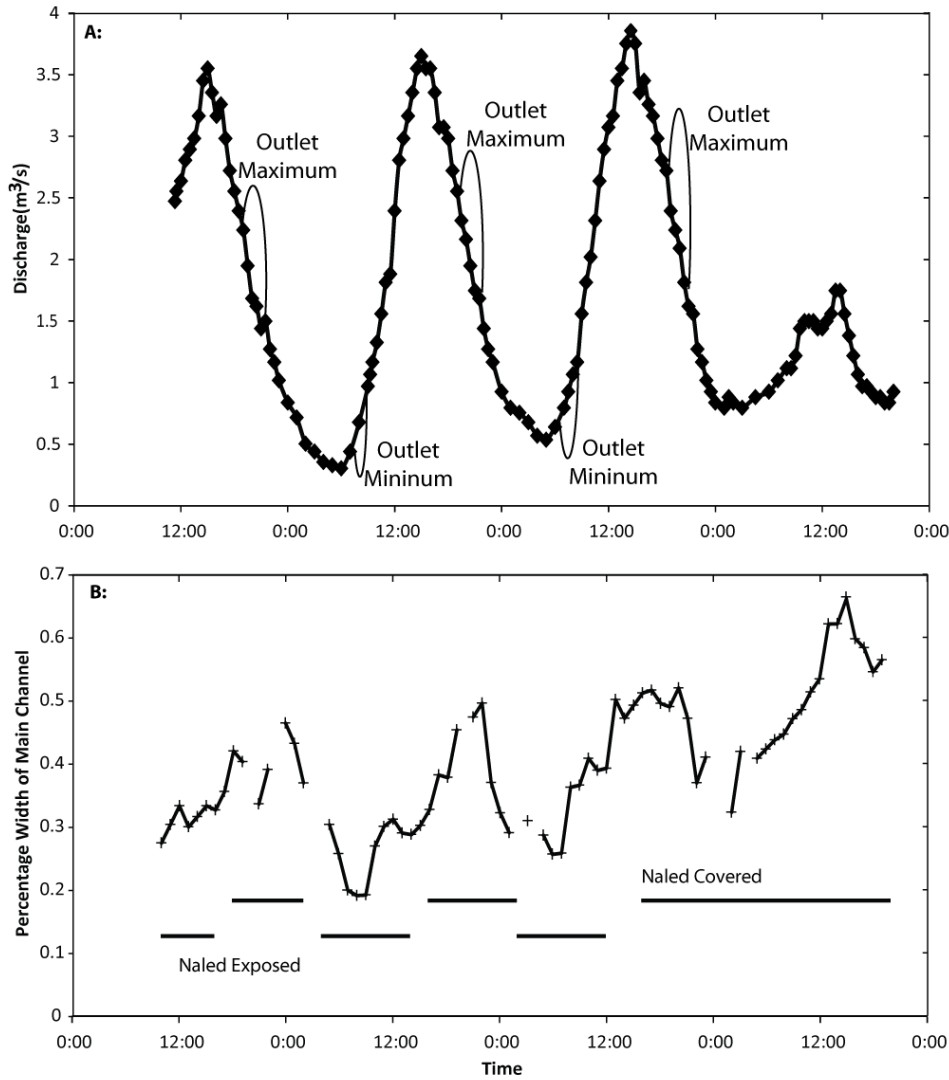


**Figure 3.** A) Measured discharge in an interior surface stream over a 4 day period in 2012 compared to
time ranges of maximum and minimum discharge as suggested by flow volumes over midstream naled
ice in the outlet of Isunnguata Sermia during the study period. B) Assessment of percentage of distance
between a point bar and the shore that is characterized by large waves suggestive of deep, fast flow.
Periods of time where the midstream naled is exposed and covered are included for comparison.



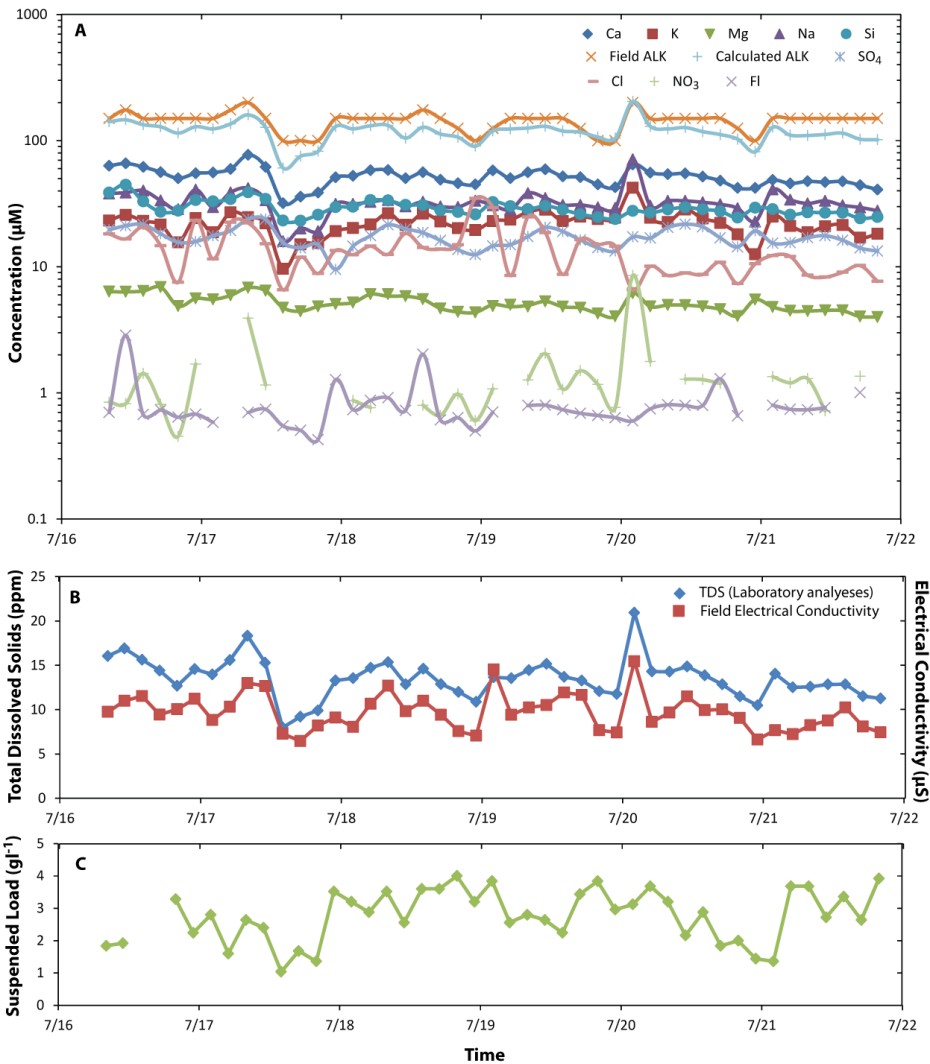

**Figure 4.** A) Concentration of dissolved constituents in sampled waters over time, including laboratory

measurements of cations and Si by ICP-MS, anions by ion chromatography, and field measurements of

alkalinity (ALK). Alkalinity as calculated by charge balance is also depicted. B) Total dissolved solids from

the sum of the laboratory measurements and charge balance alkalinity ($HCO_3$) compared to field

conductivity measurements. Co-variation is statistically significant ($p < 0.0001$). C) Dry weight of

suspended sediment on filters.



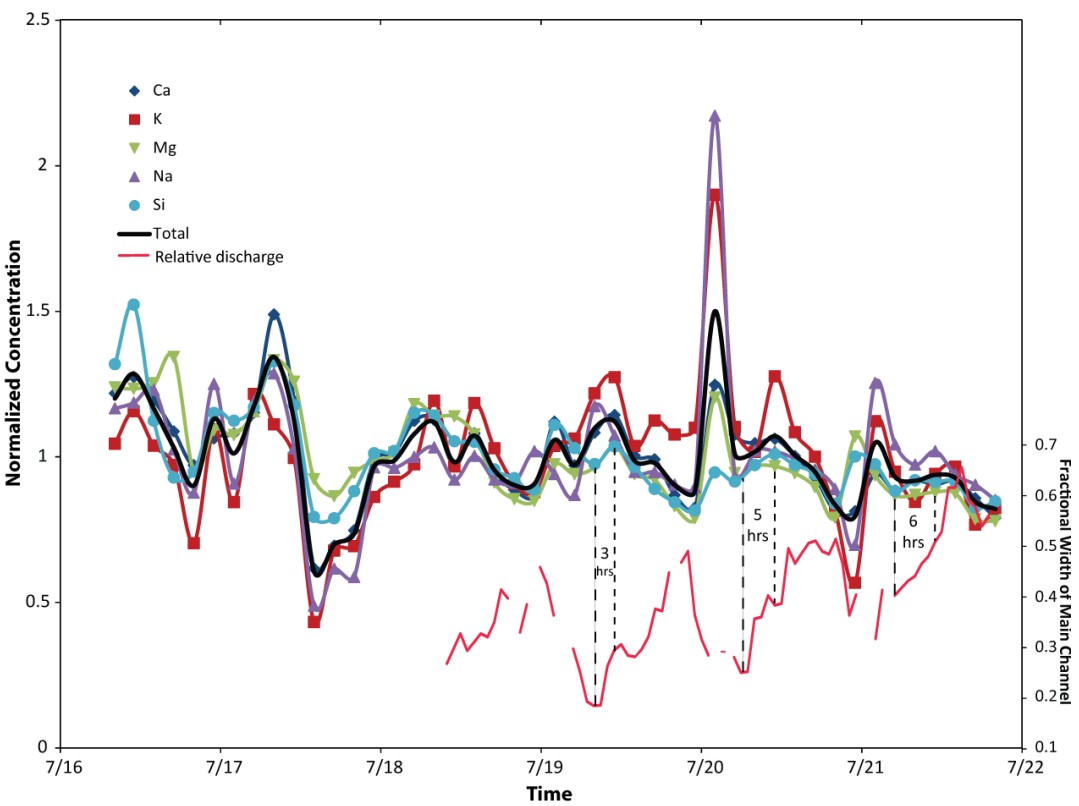

461

**Figure 5.** Concentration of dissolved cations and Si normalized to average concentration. Discharge from

relative active channel width is shown for comparison. Lags between active channel width channel

minima and solute concentration maxima are illustrated with dashed lines.

465



**Tables:**

Table 1. Field and laboratory measurements (results in micro-molarity unless otherwise noted)

| Sample | Time | pH | Eletrical Conductivity (μS) | Suspended Sediment (g l⁻¹) | Field Alkalinity (μM) | Ca | K | Mg | Na | Si | Fl | Cl | NO₃ | SO₄ | Calculated Alkalinity |
|--------|------|----|----|----|----|----|----|----|----|----|----|----|----|----|----|
| GL13-1 | 7/16/2013 8:10 | 8.59 | 9.74 | 1.84 | 150 | 63.11 | 23.24 | 6.36 | 38.12 | 38.67 | 0.70 | 18.24 | 0.85 | 19.67 | 141.17 |
| GL13-2 | 7/16/2013 11:00 | 8.82 | 10.98 | 1.92 | 175 | 65.99 | 25.72 | 6.33 | 38.77 | 44.68 | 2.86 | 16.65 | 0.82 | 21.16 | 146.46 |
| GL13-3 | 7/16/2013 14:00 | 8.70 | 11.54 | NA | 150 | 61.88 | 23.10 | 6.43 | 40.20 | 32.96 | 0.67 | 20.58 | 1.43 | 21.68 | 133.87 |
| GL13-4 | 7/16/2013 17:00 | 8.70 | 9.44 | NA | 150 | 56.32 | 21.61 | 6.89 | 33.53 | 27.25 | 0.73 | 14.71 | 0.81 | 18.27 | 128.76 |
| GL13-5 | 7/16/2013 20:00 | 8.65 | 10.04 | 3.28 | 150 | 50.31 | 15.65 | 4.87 | 28.65 | 27.84 | 0.64 | 7.53 | 0.45 | 15.63 | 114.79 |
| GL13-6 | 7/16/2013 23:00 | 8.75 | 11.22 | 2.24 | 150 | 55.07 | 24.32 | 5.60 | 40.89 | 33.75 | 0.68 | 23.38 | 1.69 | 15.88 | 129.06 |
| GL13-7 | 7/17/2013 2:00 | 8.73 | 8.82 | 2.80 | 150 | 55.96 | 18.77 | 5.52 | 29.67 | 32.96 | 0.58 | 11.53 | <0.45 | 17.62 | 124.03 |
| GL13-8 | 7/17/2013 5:00 | 8.84 | 10.32 | 1.60 | 175 | 59.65 | 27.05 | 5.94 | 38.90 | 34.34 | <0.43 | 22.57 | <0.45 | 19.35 | 135.86 |
| GL13-9 | 7/17/2013 8:00 | 8.90 | 13 | 2.64 | 200 | 77.11 | 24.73 | 6.83 | 42.06 | 38.95 | 0.70 | 22.27 | 3.91 | 23.84 | 160.13 |
| GL13-10 | 7/17/2013 11:00 | 8.51 | 12.66 | 2.40 | 150 | 61.76 | 22.14 | 6.46 | 33.64 | 34.54 | 0.74 | 15.23 | 1.16 | 23.85 | 127.40 |
| GL13-11 | 7/17/2013 14:00 | 8.01 | 7.28 | 1.04 | 100 | 31.93 | 9.62 | 4.74 | 15.96 | 23.29 | 0.55 | 6.56 | <0.45 | 15.56 | 60.69 |
| GL13-13 | 7/17/2013 17:00 | 8.32 | 6.46 | 1.68 | 100 | 35.97 | 15.07 | 4.44 | 20.12 | 23.15 | 0.51 | 11.90 | <0.45 | 14.14 | 75.32 |
| GL13-14 | 7/17/2013 20:00 | 8.62 | 8.2 | 1.36 | 100 | 38.71 | 15.41 | 4.85 | 19.19 | 25.86 | 0.43 | 8.85 | <0.45 | 14.99 | 82.45 |
| GL13-15 | 7/17/2013 23:00 | 8.80 | 9.1 | 3.52 | 150 | 51.23 | 19.17 | 5.06 | 31.51 | 29.70 | 1.27 | 13.49 | <0.45 | 9.54 | 129.42 |
| GL13-16 | 7/18/2013 2:00 | 8.71 | 8.04 | 3.20 | 150 | 52.52 | 20.33 | 5.21 | 31.44 | 29.94 | 0.74 | 12.48 | 0.87 | 14.55 | 124.02 |
| GL13-17 | 7/18/2013 5:00 | 8.53 | 10.66 | 2.88 | 150 | 58.17 | 21.66 | 6.06 | 32.69 | 33.77 | 0.87 | 14.59 | 0.76 | 17.66 | 131.26 |
| GL13-18 | 7/18/2013 8:00 | 8.90 | 12.7 | 3.52 | 150 | 58.34 | 26.52 | 5.85 | 33.74 | 33.52 | 0.92 | 12.56 | <0.45 | 21.50 | 132.18 |
| GL13-19 | 7/18/2013 11:00 | 8.48 | 9.82 | 2.56 | 150 | 50.27 | 21.52 | 5.84 | 30.12 | 30.90 | 0.73 | 18.38 | <0.45 | 19.98 | 104.80 |
| GL13-20 | 7/18/2013 14:00 | 8.53 | 10.98 | 3.60 | 175 | 56.00 | 26.34 | 5.53 | 32.80 | 30.84 | 2.03 | 14.21 | 0.80 | 18.63 | 127.88 |
| GL13-21 | 7/18/2013 17:00 | 8.66 | 9.42 | 3.60 | 150 | 48.92 | 22.88 | 4.68 | 30.15 | 28.08 | 0.61 | 13.82 | 0.66 | 16.16 | 112.80 |
| GL13-22 | 7/18/2013 20:00 | 8.61 | 7.56 | 4.00 | 125 | 45.92 | 20.21 | 4.39 | 29.79 | 27.19 | 0.64 | 14.99 | 0.98 | 13.68 | 106.64 |
| GL13-23 | 7/18/2013 23:00 | 8.60 | 7.06 | 3.20 | 100 | 44.19 | 19.53 | 4.35 | 33.33 | 26.04 | 0.50 | 34.89 | 0.60 | 12.51 | 90.43 |
| GL13-24 | 7/19/2013 2:00 | 8.68 | 14.54 | 3.84 | 125 | 58.06 | 23.07 | 5.00 | 30.77 | 32.56 | 0.71 | 29.45 | 1.08 | 14.63 | 119.46 |
| GL13-25 | 7/19/2013 5:00 | 8.27 | 9.42 | 2.56 | 150 | 50.26 | 23.64 | 4.83 | 28.40 | 30.22 | <0.43 | 8.51 | <0.45 | 14.96 | 123.76 |
| GL13-26 | 7/19/2013 8:00 | 8.58 | 10.24 | 2.80 | 150 | 56.09 | 27.09 | 4.95 | 28.60 | 29.61 | 0.79 | 24.85 | 1.27 | 17.39 | 125.83 |
| GL13-27 | 7/19/2013 11:00 | 8.70 | 10.5 | 2.64 | 150 | 59.22 | 28.32 | 5.32 | 35.07 | 30.39 | 0.80 | 18.63 | 2.05 | 20.54 | 129.90 |
| GL13-28 | 7/19/2013 14:00 | 8.46 | 11.92 | 2.24 | 150 | 52.04 | 23.07 | 4.82 | 30.99 | 28.33 | 0.74 | 8.69 | 1.07 | 19.07 | 119.13 |
| GL13-29 | 7/19/2013 17:00 | 8.35 | 11.66 | 3.44 | 125 | 51.33 | 25.01 | 4.73 | 30.98 | 26.10 | 0.69 | 16.41 | 1.50 | 16.31 | 116.88 |
| GL13-30 | 7/19/2013 20:00 | 8.58 | 7.68 | 3.84 | 100 | 44.99 | 23.94 | 4.26 | 29.57 | 24.76 | 0.66 | 15.06 | 1.17 | 14.15 | 106.81 |
| GL13-31 | 7/19/2013 23:00 | 8.58 | 7.42 | 2.96 | 100 | 42.73 | 24.46 | 4.05 | 29.26 | 23.92 | 0.64 | 14.57 | 0.77 | 13.32 | 104.67 |
| GL13-32 | 7/20/2013 2:00 | 8.53 | 15.46 | 3.12 | 200 | 64.60 | 42.27 | 6.17 | 70.99 | 27.76 | 0.60 | 6.56 | 8.54 | 17.23 | 204.64 |
| GL13-33 | 7/20/2013 5:00 | 8.03 | 8.62 | 3.68 | 150 | 55.74 | 24.52 | 4.85 | 30.50 | 26.84 | 0.75 | 10.08 | 1.78 | 16.88 | 129.83 |
| GL13-34 | 7/20/2013 8:00 | 8.37 | 9.66 | 3.20 | 150 | 54.24 | 22.61 | 4.97 | 33.37 | 28.51 | 0.81 | 8.52 | <0.45 | 20.50 | 124.06 |
| GL13-35 | 7/20/2013 11:00 | 8.30 | 11.48 | 2.16 | 150 | 54.99 | 28.39 | 4.97 | 33.30 | 29.61 | 0.79 | 8.94 | 1.29 | 21.77 | 127.04 |
| GL13-36 | 7/20/2013 14:00 | 8.39 | 9.94 | 2.88 | 150 | 51.93 | 24.12 | 4.84 | 32.49 | 28.51 | 0.79 | 8.71 | 1.28 | 20.74 | 117.89 |
| GL13-37 | 7/20/2013 17:00 | 8.48 | 10.02 | 1.84 | 150 | 48.16 | 22.25 | 4.60 | 31.28 | 27.60 | 1.29 | 10.85 | 1.19 | 16.96 | 111.80 |
| GL13-38 | 7/20/2013 20:00 | 8.69 | 8.02 | 2.00 | 125 | 42.10 | 18.12 | 4.06 | 29.09 | 24.59 | 0.66 | 7.35 | <0.45 | 14.37 | 102.76 |
| GL13-39 | 7/20/2013 23:00 | 8.12 | 6.62 | 1.44 | 100 | 42.13 | 12.60 | 5.49 | 22.80 | 29.38 | <0.43 | 10.54 | <0.45 | 19.25 | 81.60 |
| GL13-40 | 7/21/2013 2:00 | 8.80 | 7.68 | 1.36 | 150 | 48.73 | 24.96 | 4.80 | 40.99 | 28.59 | 0.80 | 12.31 | 1.35 | 15.40 | 127.78 |
| GL13-41 | 7/21/2013 5:00 | 8.20 | 7.22 | 3.68 | 150 | 45.80 | 21.11 | 4.45 | 34.07 | 25.88 | 0.74 | 12.08 | 1.20 | 15.56 | 110.55 |
| GL13-42 | 7/21/2013 8:00 | 8.70 | 8.24 | 3.68 | 150 | 47.55 | 18.79 | 4.46 | 31.80 | 26.99 | 0.74 | 8.58 | 1.30 | 17.01 | 109.97 |
| GL13-43 | 7/21/2013 11:00 | 8.42 | 8.76 | 2.72 | 150 | 46.89 | 20.93 | 4.51 | 33.33 | 26.81 | 0.76 | 8.28 | 0.72 | 17.63 | 112.03 |
| GL13-44 | 7/21/2013 14:00 | 8.12 | 10.24 | 3.36 | 150 | 47.52 | 21.48 | 4.49 | 30.72 | 26.82 | <0.43 | 9.04 | <0.45 | 16.26 | 114.65 |
| GL13-45 | 7/21/2013 17:00 | 8.32 | 8.08 | 2.64 | 150 | 44.42 | 17.05 | 4.03 | 29.55 | 24.22 | 1.01 | 10.22 | 1.36 | 14.10 | 102.73 |
| GL13-46 | 7/21/2013 20:00 | 8.62 | 7.44 | 3.92 | 150 | 40.93 | 18.31 | 4.00 | 27.84 | 24.85 | <0.43 | 7.66 | <0.45 | 13.31 | 101.73 |