# Peer review of "Combined diurnal variations of discharge and hydrochemistry of the Isunnguata Sermia"

_The Cryosphere, 2016_

## Referee Comment (RC1) · Anonymous Referee #1 · 18 Aug 2016

General overview

This is an interesting study looking at the temporal variation in the hydrochemistry of a sizeable glacial catchment in SW Greenland. On the downside, there is relatively little data – only four days of coupled relative discharge and chemical composition data, with no absolute discharge data. To be fair, this is clearly a logistically challenging site to work in, with no fixed points to allow reliable continuous data logger monitoring, and even limited robust time series data from such a large catchment is worthwhile. This paper is building on their recent Geology paper 'Chemical weathering under the

Greenland Ice Sheet', which examined the bulk chemistry of a series of boreholes waters across the catchment.

The authors do a commendable job of attempting to maximize the interpretation of the limited data set through the use of time lapse photography. The principal conclusion is that there is a phase shift between relative discharge and solute concentration, perhaps best explained by the expansion of a distributed subglacial hydrologic network into seldom accessed regions during high flow. This conclusion appears intuitively reasonable, although I have some reservations that need to be addressed. My key concern is whether the relative discharge measurements are robust enough given relatively poorly constrained potential variations in water depth and water velocity (see major points below).

Major specific points

Lines 159-162 – you base one of your major conclusions on the assumption that the width of the active channel is proportional to discharge. This is based on the assumptions that a) water velocity doesn't change, and b) water depth doesn't change (line 204). I have the following concerns/questions with this approach:

a) You state that your velocity measurements were measured only in the first two days only of the sampling period – from reading the methods, this would appear to be a period when you didn't assess discharge through photography. Since you do not have paired velocity (from surface object movement)-relative discharge measurements (from photography), how can you be sure that your conclusion that velocity does not change during the crucial last 4 day period of study is robust? (line 204; a critical underlying assumption for assuming active channel width is proportional to relative discharge). Particularly as the initial 2 day period was when the naled was completely covered in water hence discharge likely anomalously higher than remainder of study (or bedload sediment depth higher).

b) Line 159-162 – you need to give more details of your methods here. How many

repeats of surface velocity measurements were made? From results you state 6 total measurements? (this doesn't seem a lot to base your conclusions on over the whole period of study). What were the 'other objects' in the water that you use as velocity indicators? I'd also question the validity of using ice blocks as reliable relative velocity indicators – could they not become snagged by debris in river, or velocity altered by differences in wind direction and speed, position in channel etc? You also need to state that the cross sectional normalized flow rate is likely to be substantially different from the surface velocity, particularly when using non neutrally buoyant objects.

c) You also state there is no change in depth in the river, although you also state that there is considerable mobility of sediment with collapsing banks, and scouring and deposition of sediment. Surely the assumption of constant river depth (line 204) should be viewed as a guestimate?

Line 299-301 – given the large uncertainties in relative discharge measurements (see above), I don't think you can make a quantitative statement that the discharge variation is substantially larger than the variation in concentration of dissolved solutes – though fair to say that the likely width of the active channel roughly doubles.

Other specific/technical points

Line 8 – would be more correct to state that there are 4 days of continuous relative discharge and hydrochemical data

Line 11/12 – don't need to state that element and ions were measured in lab in abstract (plus you also measured suspended sed weights in lab, and

Line 17/18 – I'd omit this sentence from abstract – it is based on the prior season's discharge from a single supraglacial stream. Worth mentioning in main text, but not strong enough for abstract.

Line 56 – would be worth referencing Wadham et al Global Biogeoch Cycl 'Size Matters' article here

Line 93 – need to either reference this subjective number of 500 m3s-1 to something (visually, similar to observed discharge at x, which has a discharge of x) or omit.

Line 110 – what kind of bottle, Nalgene PP?, was it cleaned, rinsed with sample etc? how long was pole?

Line 120 – 0.1 um nylon filters seem an odd choice – most studies use 0.2 um or 0.45 um, why were these chosen? What filtration apparatus was used, Nalgene? Were procedural blanks (e.g. using MQ water) carried out, what were these blank values and how did they compare to instrument detection limits?

Line 126 – pH is always tough to accurately measure in low conductivity glacial waters. Please give further details – was 2 or 3 point calibration used, how long was pH probe left to stabilize prior to reading? Two decimal points for pH in Table 1 seems v optimistic.

Line 128 – please state precisions and detection limits here, ditto line 130

Line 130 – what temp were filters dried, and for how long?

Line 176 – typo, should be gauge

Line 176 to 178. What was the comparative weather like during the 6 day main study water sampling? To what depth into ice interior had snow line retreated to during both seasons – the outlet has a large catchment, and snow cover will have a major impact on the timing/magnitude of discharge runoff.

Line 182 – you start with discharge results. You should ideally be consistent with the order of methods section – perhaps best to start methods with discharge.

Line 200 – you need to put in these calculations, and state assumptions

Line 240 – how were lab TDS calculated: from summing inorganic ions, or by measurement using conductivity/TDS meter?

Line 343 – should put relative discharge, not just discharge – also see major points

above

Line 343 – you only measured relative discharge over 6 days, not four

Line 344-347 – given many uncertainties in measuring relative discharge, I think this statement is too strong

Table 1 – there are too many decimal points – I would have thought one decimal point is sufficient for most (though need to compare to precision of analyses). Also, need to put charges on anions to be consistent with main text. Plus better to have field alkalinity and calculated alkalinity in adjacent columns to aid comparison.

Figure 2 – these were too small in my printed copy to see properly. Please make photos larger, they are a key component of the study

Figure 3 – Rather than having smoothed lines for min/max discharge, put on hourly points (or as well)

---

## Referee Comment (RC2) · R.G. Bingham (Referee) · 16 Nov 2016

This paper provides a relatively high-resolution record of hydrochemistry measurements obtained just in front of the subglacial outlet of a western Greenland glacier over 6 days in July 2013. The record is primarily compared with discharge as assessed with time-lapse photography. The authors use these data to infer properties of the subglacial drainage system upstream from the terminus, and suggest that the lack of an inverse relationship between discharge and solute concentrations could be indicative of subglacial water accessing a linked-cavity system during peak discharge

and being effective at drawing solutes from these cavities during the falling limb.

The paper presents a useful new dataset on subglacial hydrochemistry which was clearly hard won, albeit covers rather a short period (6 days, albeit with 3 hour increments). The use of time-lapse photography to obtain a measure of relative discharge is a neat concept for overcoming the difficulties of measuring stage in such an active environment. So I think that ultimately the authors present some good material here.

However, in its current form, I did not find the discussion of the data especially insightful or even especially novel. In essence, I feel the authors have to rewrite the discussion for the paper significantly to make a convincing case that the paper is presenting a novel advance. At the moment, because the paper is based on rather limited data, I think that approach has to involve providing a far more comprehensive grounding of the ideas proposed here against what has, or they might argue has not, been interpreted from elsewhere.

My comments below concern the Discussion section (though some wider referencing and context would also benefit the introduction). I also made some minor comments throughout the paper (not including the Discussion/Conclusions) in the attached supplement.

Discussion Given the precariousness of the discharge results (I do have sympathy; I know all about the challenges of getting these data), I'd recommend the discussion explicitly focuses on the hydrochemistry variations, albeit using some of the qualitative discharge observations as context (i.e. I suggest excising Section 5.1). I then think you should partition the discussion into subsections which might broadly be described as (i) synthesise the main finding here, i.e. midsummer lag observed between hydrochemistry and discharge; and propose the conceptual model that water accesses distributed system on falling limb; (ii) compare this model comprehensively with findings/suggested interpretations of subglacial hydrological behaviour from other glacial systems where hydrology and/or hydrochemistry of meltwater have been observed. . . ..

I think the single biggest failing of the paper right now is that it doesn't adequately reference many other relevant studies, and therefore much of the context for justifying the discussion here is missing. For example, I'd say it should be well known from a number of studies of the hydrology of Greenland's outlets (e.g. from the Edinburgh and Bristol groups), and even large polythermal glaciers (Skidmore and Sharp, 1999, Annals of Glaciology) that the larger the catchment, the less likely one is to observe an "alpine-style" inverse relationship between solute concentration and discharge. Similarly, the above groups, and others, have acquired datasets that evince significant subglacial drainage system evolution as the melt season progresses many km upstream of outlet portals (e.g. Bartholomew et al., 2010, Nature Geoscience; 2011, EPSL) – and you'll see in Bingham et al. (2006; Earth Surface Proc. Landforms) evidence that by late July an Arctic subglacial system at similar latitude to your study area can be channelized, but discharge still accesses the distributed (your "linked-cavity") system at times of exceptional melt inputs. If you're going to entitle the paper "...gives insight into subglacial conditions" then I think the insight only comes by making a much more comprehensive comparison with other relevant studies.

Finally, since one of the setups of the paper is to assess whether solute/discharge follows a positive/inverse/complex relationship, a comprehensive background for this (albeit pre most Greenland hydrology studies) is given by G.H. Brown (2002) Glacier meltwater hydrochemistry, Applied Geochemistry, 17, 855-883.

Please also note the supplement to this comment:
http://www.the-cryosphere-discuss.net/tc-2016-137/tc-2016-137-RC2-supplement.pdf

---

## Author Comment (AC3) · 24 Jan 2017

Below we reply to reviewer line by line. We have marked the paragraphs of the original review with "«" and "»" symbols.

«General overview This is an interesting study looking at the temporal variation in the hydrochemistry of a sizeable glacial catchment in SW Greenland. On the downside, there is relatively little data – only four days of coupled relative discharge and chemical composition data, with no absolute discharge data. To be fair, this is clearly a logistically challenging site to work in, with no fixed points to allow reliable continuous data logger

monitoring, and even limited robust time series data from such a large catchment is worthwhile. This paper is building on their recent Geology paper 'Chemical weathering under the Greenland Ice Sheet', which examined the bulk chemistry of a series of boreholes waters across the catchment.»

We thank the reviewer for their interest in our study. We felt that although the data record presented is relatively short, it is still of interest to the broader community, both in that it presents novel methods for assessing relative discharge and in the unusual features of the record that are observed.

«The authors do a commendable job of attempting to maximize the interpretation of the limited data set through the use of time lapse photography. The principal conclusion is that there is a phase shift between relative discharge and solute concentration, perhaps best explained by the expansion of a distributed subglacial hydrologic network into seldom accessed regions during high flow. This conclusion appears intuitively reasonable, although I have some reservations that need to be addressed. My key concern is whether the relative discharge measurements are robust enough given relatively poorly constrained potential variations in water depth and water velocity (see major points below).»

Some of the objections in the specific major points section (below) are valid and we have addressed the corresponding uncertainties in the revised version of the manuscript. However, we feel strongly that our photographic record is robust in assessing relative discharge. The reviewer's main objections focus on the relatively few and imprecise velocity measurements. We agree that it would have been wise to assess velocity more frequently. However, the doubling of the width of the active change over the course of a day is a more important signal than velocity. A well-established property of braided streams in unconsolidated fluvial sediment is that they expand and contract. Without changes in depth or gradient, changes in velocity are not possible. Furthermore, if there were velocity changes that we failed to observe, we would expect the higher flow periods to have higher velocities. This would accentuate the change in
relative discharge that we observed, not diminish it.

«Major specific points Lines 159-162 – you base one of your major conclusions on the assumption that the width of the active channel is proportional to discharge. This is based on the assumptions that a) water velocity doesn't change, and b) water depth doesn't change (line 204). I have the following concerns/questions with this approach: a) You state that your velocity measurements were measured only in the first two days only of the sampling period – from reading the methods, this would appear to be a period when you didn't assess discharge through photography. Since you do not have paired velocity (from surface object movement)-relative discharge measurements (from photography), how can you be sure that your conclusion that velocity does not change during the crucial last 4 day period of study is robust? (line 204; a critical underlying assumption for assuming active channel width is proportional to relative discharge).»

Though we did not systematically photograph the stream from a consistent vantage point during the first two days of the study, we did still take plenty of photographs and field notes about the stream's behavior. These qualitative observations support that conclusion that the stream's behavior is consistent between the first two and later four days of the study. I.e. the naled is exposed and covered, and the width of the active channel grows and shrinks similarly over both periods. We have modified both the methods and results to discuss this qualitative similarity between the periods in the text. (Lines 141 and 231)

«Particularly as the initial 2 day period was when the naled was completely covered in water hence discharge likely anomalously higher than remainder of study (or bedload sediment depth higher).»

The reviewer's impression that the naled was covered for the entire first two days of the study is incorrect. (I do not know what portion of the text gave that impression.)

«b) Line 159-162 – you need to give more details of your methods here. How many repeats of surface velocity measurements were made? From results you state 6 total

measurements? (this doesn't seem a lot to base your conclusions on over the whole period of study). What were the 'other objects' in the water that you use as velocity indicators? I'd also question the validity of using ice blocks as reliable relative velocity indicators – could they not become snagged by debris in river, or velocity altered by differences in wind direction and speed, position in channel etc? You also need to state that the cross sectional normalized flow rate is likely to be substantially different from the surface velocity, particularly when using non neutrally buoyant objects.»

We have now described our method of timing the stream's surface flow in greater detail (line 136).

«c) You also state there is no change in depth in the river, although you also state that there is considerable mobility of sediment with collapsing banks, and scouring and deposition of sediment. Surely the assumption of constant river depth (line 204) should be viewed as a guestimate?»

We have rewritten the language to better reflect the uncertainty (line 220)

«Line 299-301 – given the large uncertainties in relative discharge measurements (see above), I don't think you can make a quantitative statement that the discharge variation is substantially larger than the variation in concentration of dissolved solutes – though fair to say that the likely width of the active channel roughly doubles.»

This statement has been removed.

«Other specific/technical points Line 8 – would be more correct to state that there are 4 days of continuous relative discharge and hydrochemical data»

Text has now been added to clarify that only four of those days had measurements based on repeat photography. During the first two days, we attempted to use velocity and a stage pole. This was unsuccessful for reasons discussed on lines 114-115.

«Line 11/12 – don't need to state that element and ions were measured in lab in abstract (plus you also measured suspended sed weights in lab, and»

All measurements employed in the study now appear on a single list (without discussion of which are lab and which are field).

«Line 17/18 – I'd omit this sentence from abstract – it is based on the prior season's discharge from a single supraglacial stream. Worth mentioning in main text, but not strong enough for abstract.»

The sentence has been removed.

«Line 56 – would be worth referencing Wadham et al Global Biogeoch Cycl 'Size Matters' article here»

The Wadham et al. citation is appropriate here and has been added.

«Line 93 – need to either reference this subjective number of 500 m3s-1 to something (visually, similar to observed discharge at x, which has a discharge of x) or omit.»

We have modified the statement to be more qualitative and added referenced the quantitative measurements made in the Watson River.

«Line 110 – what kind of bottle, Nalgene PP?, was it cleaned, rinsed with sample etc? how long was pole?»

More detail about the bottle, pole, and sample rinsing is now included (Line 169)

«Line 120 – 0.1 um nylon filters seem an odd choice – most studies use 0.2 um or 0.45 um, why were these chosen? What filtration apparatus was used, Nalgene? Were procedural blanks (e.g. using MQ water) carried out, what were these blank values and how did they compare to instrument detection limits?»

Our choice to use such fine filters came from concern that colloidal sized particles (from glacial commutation) were able to pass through larger filters. However, we feel that a discussion of this point is beyond the scope of the paper. We now discuss procedural blanks (line 191).

«Line 126 – pH is always tough to accurately measure in low conductivity glacial waters. Please give further details – was 2 or 3 point calibration used, how long was pH probe left to stabilize prior to reading? Two decimal points for pH in Table 1 seems v optimistic.»

This is now discussed in lines 186.

«Line 128 – please state precisions and detection limits here, ditto line 130»

Does the reviewer intend for us to list the lower limits of detection for all analyzed elements in the manuscript text? I don't believe this is customary.

«Line 130 – what temp were filters dried, and for how long?»

This is now in the manuscript (line 192)

«Line 176 – typo, should be gauge»

Fixed.

«Line 176 to 178. What was the comparative weather like during the 6 day main study water sampling? To what depth into ice interior had snow line retreated to during both seasons – the outlet has a large catchment, and snow cover will have a major impact on the timing/magnitude of discharge runoff.»

We now discuss the temperature during the study period on lines 166-168. We discuss the comparability of the ablation seasons in lines 150-152. As it turns out, the total progress of ablation was comparable between the two field seasons. The 2012 data are from earlier in the season, but from a year of extraordinary melt.

«Line 182 – you start with discharge results. You should ideally be consistent with the order of methods section – perhaps best to start methods with discharge.»

We have reordered the methods section to mirror the results section.

«Line 200 – you need to put in these calculations, and state assumptions»

We added a parenthetical explaining the basic concept (line 216). We feel that the equation sufficiently well known that further elaboration is not necessary.

«Line 240 – how were lab TDS calculated: from summing inorganic ions, or by measurement using conductivity/TDS meter?»

The text now reads "sum of laboratory measured inorganic ions."

«Line 343 – should put relative discharge, not just discharge – also see major points»

The word "relative" has been added.

«Line 343 – you only measured relative discharge over 6 days, not four»

The reference to six days has been removed from the sentence.

«Line 344-347 – given many uncertainties in measuring relative discharge, I think this statement is too strong»

We have replaced the word "show" with "suggest" to indicate greater uncertainty.

«Table 1 – there are too many decimal points – I would have thought one decimal point is sufficient for most (though need to compare to precision of analyses). Also, need to put charges on anions to be consistent with main text. Plus better to have field alkalinity and calculated alkalinity in adjacent columns to aid comparison.»

We have dropped the second decimal on everything but the suspended sediment analysis and made the other requested changes.

«Figure 2 – these were too small in my printed copy to see properly. Please make photos larger, they are a key component of the study»

We have increased the size of the figure and zoomed in most of the images to enlarge the most relevant details.

«Figure 3 – Rather than having smoothed lines for min/max discharge, put on hourly points (or as well)»

It seemed inappropriate to actually label figure 3a with hourly points from figure 3b, as the two datasets are from different time periods. The smoothed max-min curves are meant to represent the range suggested by the combination of qualitative and quantitative observations discussed in the text. The figure caption has been modified to reflect this.

---

## Author Comment (AC4) · 24 Jan 2017

Please see our line by line reply below. The original review is marked with the "«" ad "»" symbols.

«This paper provides a relatively high-resolution record of hydrochemistry measurements obtained just in front of the subglacial outlet of a western Greenland glacier over 6 days in July 2013. The record is primarily compared with discharge as assessed with time-lapse photography. The authors use these data to infer properties of the subglacial drainage system upstream from the terminus, and suggest that the lack of an

inverse relationship between discharge and solute concentrations could be indicative of subglacial water accessing a linked-cavity system during peak discharge and being effective at drawing solutes from these cavities during the falling limb. The paper presents a useful new dataset on subglacial hydrochemistry which was clearly hard won, albeit covers rather a short period (6 days, albeit with 3 hour increments). The use of time-lapse photography to obtain a measure of relative discharge is a neat concept for overcoming the difficulties of measuring stage in such an active environment. So I think that ultimately the authors present some good material here.»

We thank Dr. Bingham for his interest in our data set.

«However, in its current form, I did not find the discussion of the data especially insightful or even especially novel. In essence, I feel the authors have to rewrite the discussion for the paper significantly to make a convincing case that the paper is presenting a novel advance. At the moment, because the paper is based on rather limited data, I think that approach has to involve providing a far more comprehensive grounding of the ideas proposed here against what has, or they might argue has not, been interpreted from elsewhere. My comments below concern the Discussion section (though some wider referencing and context would also benefit the introduction). I also made some minor comments throughout the paper (not including the Discussion/Conclusions) in the attached supplement.»

We have revised the introduction and discussion to better reference the wide range of contexts in which hysteresis between solute flux and discharge is observed, including non-glacial settings. Whereas gradual increases solute flux during waxing flow may been observed in a wide range of contexts, we feel that the observed spikes during waning flow are in fact novel. The only previous study that (to our knowledge) has reported such behavior in a glacial context is the Anderson and others paper cited in the introduction. And in that study, the phenomenon occurred on a multi-day timescale, whereas it occurred on an hourly scale here.

«Discussion Given the precariousness of the discharge results (I do have sympathy; I know all about the challenges of getting these data), I'd recommend the discussion explicitly focuses on the hydrochemistry variations, albeit using some of the qualitative discharge observations as context (i.e. I suggest excising Section 5.1). I then think you should partition the discussion into subsections which might broadly be described as (i) synthesise the main finding here, i.e. midsummer lag observed between hydrochemistry and discharge; and propose the conceptual model that water accesses distributed system on falling limb; (ii) compare this model comprehensively with findings/suggested interpretations of subglacial hydrological behaviour from other glacial systems where hydrology and/or hydrochemistry of meltwater have been observed.»

We have rewritten the first paragraph of section 5.2 to discuss differences and similarities with other hydrological systems. We felt it was necessary to keep section 5.1 mostly because the Smith and others paper argued against any diurnal changes in discharge at Isunnguata Sermia, and our study refutes that.

«I think the single biggest failing of the paper right now is that it doesn't adequately reference many other relevant studies, and therefore much of the context for justifying the discussion here is missing. For example, I'd say it should be well known from a number of studies of the hydrology of Greenland's outlets (e.g. from the Edinburgh and Bristol groups), and even large polythermal glaciers (Skidmore and Sharp, 1999, Annals of Glaciology) that the larger the catchment, the less likely one is to observe an "alpine-style" inverse relationship between solute concentration and discharge. Similarly, the above groups, and others, have acquired datasets that evince significant subglacial drainage system evolution as the melt season progresses many km upstream of outlet portals (e.g. Bartholomew et al., 2010, Nature Geoscience; 2011, EPSL) – and you'll see in Bingham et al. (2006; Earth Surface Proc. Landforms) evidence that by late July an Arctic subglacial system at similar latitude to your study area can be channelized, but discharge still accesses the distributed (your "linked-cavity") system at times of exceptional melt inputs. If you're going to entitle the paper "gives insight into subglacial

conditions" then I think the insight only comes by making a much more comprehensive comparison with other relevant studies.»

We now include the papers you suggest in our discussion section.

«Finally, since one of the setups of the paper is to assess whether solute/discharge follows a positive/inverse/complex relationship, a comprehensive background for this (albeit pre most Greenland hydrology studies) is given by G.H. Brown (2002) Glacier meltwater hydrochemistry, Applied Geochemistry, 17, 855-883.»

The Brown paper is now cited in the introduction.

«Please also note the supplement to this comment: http://www.the-cryosphere-discuss.net/tc-2016-137/tc-2016-137-RC2-supplement.pdf Interactive comment on The Cryosphere Discuss., doi:10.5194/tc-2016-137»

Most of the annotations on the PDF have be implemented as requested.

---

## Author Response (AR2)

Dear Rob Bingham,

We have implemented your final technical corrections. The line item corrections were implemented as requested with minimal changes to the manuscript. The requested changes to the date format was implemented throughout the manuscript and in table 1, figure 4, and figure 5. The requested changes are italics below:

*Line 78: remove "exists" (erroneous second instance in this sentence)*
*Line 87: lowercase, "western"*
*Lines 284-294: Would be improved if all sentences were written consistently in past tense.*
*Line 361: sp. Glen (one n)*
*Throughout manuscript: I would suggest formatting all dates as e.g. 18 July 2013, rather than e.g. 07/13/2013. The former format is more intuitive to the international audience.*

Sincerely,

Joseph Graly